# An Insight into Durability, Electrical Properties and Thermal Behavior of Cementitious Materials Engineered with Graphene Oxide: Does the Oxidation Degree Matter?

**DOI:** 10.3390/nano13040726

**Published:** 2023-02-14

**Authors:** Francesca Romana Lamastra, Giampiero Montesperelli, Emanuele Galvanetto, Mehdi Chougan, Seyed Hamidreza Ghaffar, Mazen J. Al-Kheetan, Alessandra Bianco

**Affiliations:** 1Dipartimento di Ingegneria dell’Impresa “Mario Lucertini” and Consorzio INSTM Unità di Ricerca “Roma Tor Vergata”, Università degli Studi di Roma “Tor Vergata”, Via del Politecnico, 00133 Roma, Italy; 2Dipartimento di Ingegneria Industriale (DIEF), Università di Firenze, Via di Santa Marta 3, 50139 Firenze, Italy; 3Department of Civil and Environmental Engineering, Brunel University London, Uxbridge UB8 3PH, Middlesex, UK; 4Applied Science Research Center, Applied Science Private University, Jordan; 5Department of Civil and Environmental Engineering, College of Engineering, Mutah University, Mutah 61710, Karak, Jordan

**Keywords:** graphene oxide, nanocomposites, electrical resistivity, thermal conductivity, transport properties, porosity

## Abstract

Due to global environmental concerns related to climate change, the need to improve the service life of structures and infrastructures is imminently urgent. Structural elements typically suffer service life reductions, leading to poor environmental sustainability and high maintenance costs. Graphene oxide nanosheets (GONSs) effectively dispersed in a cement matrix can promote hydration, refine the microstructure and improve interfacial bonding, leading to enhanced building materials’ performance, including mechanical strength and transport properties. Cement-based nanocomposites engineered with GONSs were obtained using two commercial nanofillers, a GO water suspension and a free-flowing GO nanopowder, characterized by fully comparable morphology, size and aspect ratio and different oxidation degrees (i.e., oxygen-to-carbon molar ratio), 0.55 and 0.45, respectively. The dosage of the 2D-nanofiller ranged between 0.01% and 0.2% by weight of cement. The electrical and thermal properties were assessed through electrochemical impedance spectroscopy (EIS) and a heat flow meter, respectively. The results were discussed and linked to micrometric porosity investigated by micro-computed tomography (μ-CT) and transport properties as determined by initial surface absorption test (ISAT), boil-water saturation method (BWS) and chloride ion penetration test. Extra-low dosage mortars, especially those loaded with a lower oxidation degree (i.e., 0.45GO), showed decreased permeability and improved barrier to chloride ion transport combined with enhanced thermal and electrical conductivity with respect to that of the control samples.

## 1. Introduction

The construction sector consumes 40% of the world’s raw materials, generating greenhouse gas (GHG) emissions and acid rain agents. The production of Portland cement for concrete manufacturing, the most heavily consumed material in the sector, accounts for about 7% of global GHG emissions [1]. The urgent need to minimize the negative environmental impacts of cement manufacturing and production motivates researchers to develop new cementitious materials that are environmentally friendly and more cost-effective than traditional materials. The durability of cementitious materials is mainly affected by their ability to absorb water, chloride ions, acids, alkalis, CO_2,_ or other aggressive substances, which induce degradation mechanisms that might cause severe damage and structural deterioration. In this context, the effort of scientists is currently focused on developing reliable and cost-effective strategies to improve the durability of construction materials. In fact, extended durability is beneficial for both service life and GHG emissions recovery [2,3]. Advanced cementitious nanocomposites based on cutting-edge nanomaterials, mainly nano-silica and graphene nanosheets and derivatives (GNDs), are among the most promising and competitive 21st-century construction materials. Moreover, commonly employed fiber-reinforced cementitious materials typically suffer from limitations inherent to structural applications on a full scale, including low modulus of elasticity, high production costs and high cement dosage [4,5,6,7]. In this framework, the investigation of cementitious materials (cement paste, cement mortar, ordinary concrete, high-performance concrete) and geopolymers engineered with graphene oxide nanosheets (GONSs) is a trending topic. In 2022, an accurate screening performed by a bibliometric analysis based on co-occurrence methodology revealed an increasing publication trend over the last decade (2011–2021). The origin of such interest, mainly focused on the correlation between microstructural features and mechanical properties, the performance of GONSs achieved at the nanoscale due to their ability and efficiency to interact with the binder associated with high specific surface area and strong covalent bond with cement hydration products [8,9]. The durability improvement of GONS-modified cementitious nanocomposites related to pore structure refinement results in improved transport properties (i.e., water permeability, gas permeability, chloride penetration resistance), erosion resistance, freeze–thaw resistance and carbonation resistance, which attracts growing attention worldwide [10,11]. Quickly increasing interest in the research community is also devoted to understanding GONS-modified cementitious nanocomposites’ electrical and thermal behavior in view of their potential applications. The electrical characterization of cementitious composites provides a practical insight into the microstructural features of GONS-based nanocomposites. Furthermore, the resistance to electrical charges is governed by the complex combination of GONSs-dependent variables, mainly dosage, size, oxidation degree, and shape factor, that diversely induce reduced availability of free water, increased density, pore refining, accelerated hydration kinetics and excess of hydration products throughout the capillary pore network [8,9]. Regarding the thermal behavior of cementitious composites, it is worth mentioning that in modern concrete technology, the control of thermal cracks is still challenging, with the great temperature gradient between the surface and the core mainly responsible for generating these defects. Recently, G. Jing et al. [12] demonstrated that cementitious composites loaded with GNDs display a reduction in thermal stress, which restrains the development of thermal cracks. From this perspective, the aim of this investigation is to contribute to the knowledge of durability and electrical and thermal behavior of GONS-modified cementitious materials [9] with a particular focus on the impact of the oxidation degree on these properties. For this purpose, two commercially available GONSs characterized by 0.55 and 0.45 oxidation degrees were employed [13,14]. The dosage ranged between 0.01% (by weight of cement), considered an extra-low value, and 0.2% (by weight of cement). The samples were hardened in water at room temperature (RT) for 7, 14 and 28 days. Based on our knowledge, little attention has been paid by researchers to the transport properties, thermal and electrical conductivity of cementitious nanocomposites with various oxidation degree of GONSs.

## 2. Materials and Methods

### 2.1. Materials and Fabrication of GO-Engineered Nanocomposites

Graphene oxide free-flowing nanopowder (*GO*) (Sigma-Aldrich, Milano, Italy) and Graphene oxide water suspension (*GONan*) (GONan, Nanesa, Roma, Italy) were employed. A systematic characterization of the two nanofillers, having fully comparable morphology, size and aspect ratio, has been previously reported [13,14]. The characterization of the two GOs by infra-red and Raman spectroscopy also indicates the presence in both nanofillers of the same oxygen-containing functional groups (i.e., hydroxyl, carbonyl, carboxyl and epoxide groups) and a higher number of defects and oxygen content in *GONan* with respect to *GO* [13,14]. The selected data are resumed in Table 1.

Nano-engineered cementitious composites have been prepared without additives using a commercially available premixed dry mortar (class M5) that, according to EN 998-2 standard, contains 1 part Portland Cement Type (I), 0.75 parts of hydrated lime, and 2 parts of graded sand. Briefly, tap water was added to the dry mortar, and the mixture was mechanically stirred at 500 rpm. For nanocomposite samples, the as-received (i.e., *GONan*) or prepared (i.e., *GO*) aqueous suspension was previously added, and the “contained” water computed within the total requested amount. All the details on the experimental procedures of sample preparation and characterization techniques of both fresh mixtures and hardened samples have been reported in previous research by Chougan M. et al. [13,14]. The formulation and the designation of the herein-investigated GO-engineered nanocomposites are presented in Table 2. The microstructure of all the GO-engineered mortars (Table 2) was previously investigated by scanning electron microscopy (SEM) [13,14]. It is worth mentioning that for common GO dosages (i.e., 0.02–0.04% by weight of cement), the dispersion of the nanofiller cannot be unambiguously assessed by SEM mainly due to the low content and the poor contrast between the nanofiller and the hydrate cement phase. In fact, according to the literature, large GO agglomerates have been detected only in sample 0.55GO-0.2 [13,14,15,16]. Then, regarding the dispersion of GO within a cementitious matrix, the relative improvement of mechanical and physical properties is usually considered reliable feedback that an appropriate dispersion degree has been achieved [16,17,18,19].

### 2.2. Formulation and Designation of GO-Engineered Nanocomposites

The volume fraction percentage of the nanofiller (ϕ_v_) was calculated considering the 2D platelets as rectangular solid fibers of average width (W), length (L) and thickness (t), as follows [20]:(1)$$\phi_{V}=\frac{W_{P}}{W_{P}+\left(\frac{d_{\mathrm{GO}}}{d_{m}}\right)\left(1-W_{P}\right)}$$
where W_p_ is the weight fraction percentage of GO; d_GO_ is the density of graphene oxide and d_m_ is the density of the matrix. The d_GO_ can be estimated by scaling the density of fully dense graphite (density 2.25 g/cm^3^) with t′~0.34 nm. Following this approach, based on data in Table 1, d_GO_ is approximately 0.976 g/cm^3^ and 0.981 g/cm^3^ for 0.45GO and 0.55GO, respectively [20]. The resulting volume fraction percentages of graphene oxide (ϕ_v_) are reported in Table 2. Notably, for these multiphase porous materials, the d_m_ value depends on several parameters, including temperature, curing time, amount of water, type and amount of nanofiller and, eventually, additives. Then, selecting the most suitable value to be used in Equation (1) is not straightforward and obvious. The authors chose to estimate the ϕ_v_ referring to the density of the control sample, thus considered as the matrix. Alternatively, ϕ_v_ can be merely calculated as the ratio between the volume corresponding to the actual weight of the nanofiller and the 4 × 4 × 16 cm bar volume (i.e., 256 cm^3^). Interestingly, according to the data reported in Table 2, comparable results were obtained.

### 2.3. Electrochemical Characterization of GO-Engineered Nanocomposites

All GO-modified mortars (Table 2) cured in water for 7, 14 and 28 days were analyzed by electrochemical impedance spectroscopy (EIS) (VMP3, BioLogic Science Instruments, Seyssinet-Pariset, France) on 160 × 40 × 40 mm^3^ bars. Measurements were collected at room temperature at 100% relative humidity (RH) by applying an alternate signal (amplitude 20 mV and frequency range 10 mHz–100 kHz) using a uniaxial two-point electrode method [21]. There is no general specification on the optimum frequency since it dramatically varies with moisture content and mix design; usually, the upper limit is at least 10 kHz [22]. A wet sponge was placed between the sample’s outer surface and the copper electrodes to ensure complete contact. The impedance spectra were analyzed through the equivalent circuits by means of Zview 4.0 software (Scribner Associates Inc., Charlottesville, VA, USA) based on the non-linear least squares method. To consider the deviation from ideal behavior, the equivalent circuits included resistive elements and constant phase elements (CPE) to simulate a real capacitor. The CPE expression is:(2)$$Z_{\mathrm{CPE}}=\frac{1}{Q{\left(j\omega \right)}^{n}}$$
where $j=\sqrt{-1}$, Q is the pseudo-capacitative coefficient, ω is the angular frequency and n can assume values between 0 and 1 (n = 0 represents the pure resistor, and n = 1 represents the perfect capacitor). The CPE parameters were converted into capacitance values through the equation:(3)$$C=\frac{{\left(Q\cdot R\right)}^{\frac{1}{n}}}{R}$$
where R is the resistance in parallel with CPE [23].

The resistivity (ρ) of the cement-based nanocomposites was calculated from the highest magnitude value of impedance spectra in the Bode representation, employing Equation (4) [21]:(4)$$\rho =\frac{R\cdot S}{L}$$
where R is the sample resistance, S is the sample’s cross-section and L is the distance between the two electrodes. It is well known that a slight change in the saturation level affects a mortar’s conductivity (or resistivity) as it leads to variation in the amount of water trapped in the porous network. In order to obtain reliable and repeatable measurements, EIS was then performed on samples in a saturated surface dry (SSD) condition [21]. Statistical evaluation was carried out utilizing Minitab^®^ 14 statistical software (State College, PA, USA) to validate the accuracy of the output data with standard deviation.

### 2.4. Micro-Computed Tomography (μ-CT) Analysis of GO-Engineered Nanocomposites

Micro-computed tomography (μ-CT) analysis was performed on a made-on-purpose 20 mm-edge cubic specimen (Table 2) [21]. The samples were analyzed by collecting μ-CT data using a Skyscan 1172 high-resolution micro-CT (Bruker, Billerica, MA, USA) with a microfocus (5 μm focal spot size) tungsten X-ray tube. Each sample was placed on a pedestal between the X-ray tube source and the charge-coupled device detector. A resolution of 17.1854 μm in terms of pixel size, with acquisition conditions of 100 kV, 100 μA, rotation step of 0.35°, Cu+Al filter and exposure time of 2655 ms were used, giving a total acquisition time of approximately 2 h and 25 min for each sample. The 3D image of the object’s internal structure, consisting of about 1030 slices, was reconstructed using *NRecon* v1.7.4.6 software (Bruker, Billerica, MA, USA) based on a modified Feldkamp algorithm for cone-beam acquisition geometry, with alignment and beam hardening corrections made before starting the reconstruction process. *CT-*Analyser (CTan v1.20.8.0, Bruker, Billerica, MA, USA) software was used for image clean-up, filtering and measurements. Optimized custom processing by semi-automated procedures (noise reduction by Gaussian blur in 3D space and thresholding) was used to select regions of interest (ROIs) of the samples obtained by fitting exactly (shrink-wrapped boundary) the whole volume. Different levels of image elaborations were checked to assess the same trends in the results, and in order to avoid artifacts and data misleading, CTVox v.3.3.1 and CTVol v.2.3.2.0 programs (Bruker, Billerica, MA, USA) were used for 3D visualization of the microstructure [24].

### 2.5. Permeability Tests of GO-Engineered Nanocomposites

The initial surface absorption test (ISAT) was performed according to BS 1881-208:1996. A batch of three cubic specimens 100 mm × 100 mm **×** 100 mm was considered. Previously, samples were dried at 105 ± 5 °C for 24 h, and a 200 cm^2^ circular cap sealed the top surface. The penetration of deionized water through the sample’s top surface was allowed for 10, 30, and 60 min. The ISAT rate was measured according to the following Equation (5) [25]:(5)$$f=\frac{60}{t}\times D\times 0.01$$
where f is the initial surface absorption rate (mL/m^2^·s), D is the number of scale divisions during the test and t is testing time (s).

The boil-water saturation method (BWS) was conducted on a batch of three specimens of size 40 mm × 40 mm × 160 mm, as per the ASTM C 642, to evaluate the permeable porosity. Each specimen was weighed after (i) 48 h of oven-drying at 110 °C, (ii) 48 h of submerging in tap water and finally, (iii) 5 h of immersing in boiling water followed by water cooling to determine the oven-dry mass, saturated mass after immersion and saturated mass after boiling, respectively. Based on the aforementioned data, the volume of permeable voids (VPV) was calculated according to Equation (6) [25,26]:(6)VPV (%)=[(C−A)/(C−D)]×100
where A (g) is the mass of the oven-dried sample, B (g) is the mass of the surface-dry sample after immersion, C (g) is the mass of the surface-dry sample after immersion and boiling, D (g) is apparent mass in water after immersion and boiling.

Chloride ion penetration was assessed using the salt ponding test following the BS 14629:2007 procedure with some modifications. A batch of three 100 mm × 100 mm × 100 mm cubic specimens was previously oven-dried at 110 °C for 24 h and then exposed to a 5 wt.% sodium chloride water solution [25]. After 30 days of exposure followed by 24 h of drying at 110 °C, the samples were drilled to 5, 10, 15, 20 and 25 mm depths, and the resulting powders were collected. Following Volhard’s method, silver nitrate water solution (0.02 M) was employed as the titration agent, and chloride ion content (CC) was determined according to the following Equation (7) [27,28]:(7)CC (%)=3.545×F×(V2−V1)/M
where F is the molarity of the silver nitrate solution, V2 is the ammonium thiocyanate solution volume used in the blank titration (mL), V1 is the ammonium thiocyanate solution volume used in the titration (mL) and M is the sample mass (g).

### 2.6. Thermal Conductivity of GO-Engineered Nanocomposites

Thermal conductivity measurements were carried out on 100 mm × 100 mm × 20 mm samples using a heat flow meter (Lasercomp Fox 200, TA Instruments, New Castle, DE, USA) coupled with WinTherm32 software, version 3 (TA Instruments, New Castle, DE, USA) [25]. In order to determine the thermal conductivity, all the specimens were placed between hot and cold plates with temperatures of 20 °C and 0 °C, respectively.

## 3. Results and Discussion

### 3.1. Resistivity of GO-Engineered Mortars

The EIS spectra of cement-based samples are influenced by many parameters such as porosity, pore solution, hydration products and unreacted cement. For this reason, several models have been proposed to develop equivalent circuits to analyze EIS data [29,30,31]. One of the most accepted models is the one proposed by Song G. et al. in 2000 [30], which considers the contribution of conductive paths of continuous and discontinuous pores to cement conductivity. The resulting equivalent circuit involves the resistance values of the continuous conductive paths (R_ccp_) and the resistance and capacity of the discontinuous conductive paths (R_cp_ and C_cp_) together with the capacity of the cement matrix (C_mat_). Considering that C_mat_ usually shows extremely low values that cannot be easily measured in the usual frequency ranges, it can be neglected. Therefore, the EIS spectra of cementitious samples can be reasonably described by the equivalent circuit shown in Figure 1, in which R_S_ and R_1_ allow to calculate R_ccp_ and R_cp_ through the relations:(8)$$R_{CP}=\frac{\left(R_{S}+R_{1}\right)R_{S}}{R_{1}}$$
(9)$$R_{CCP}=\left(R_{S}+R_{1}\right)$$

R_S_ and R_1_ were calculated from the intercepts of the semicircle of the fitting with the real axis (Z′), as shown in Figure 2. Since conduction of cement-based materials is ionic, the copper electrode acts as a blocking electrode. Therefore, a large semicircle at low frequencies relative to electrode behavior was observed in all spectra. However, this semicircle was neglected in the spectra analysis as it was not significant. Results of EIS measurements in terms of fitting parameters (R_S_, R_1_ and R_ccp_*)* and resistivity (ρ) are reported in Table 3, the equivalent circuit is reported in Figure 1 and selected fitted EIS data are presented in Figure 2, Figure 3 and Figure 4.

According to data reported in Table 3, the resistivity of 0.45GO-0.01 progressively increased with the curing time and followed the same trend observed for the control sample. On the other hand, all 0.55GO-modified samples showed a different evolution of resistivity characterized by a maximum value for 14 days cured specimen progressively increasing with the GO dosage. Moreover, results in Table 3 revealed that despite the different oxidation degree and/or dosage of the employed GO (Table 1), at 28 days, all samples except 0.55GO-0.2 exhibited decreased resistivity within 34% and 37% compared to that of the control sample, independently from GO oxidation degree.

The cement matrix’s conductive mechanism is associated with ionic conductivity due to ion migration through the network of capillary pores. The electrical properties of GNDs/cement composites are driven by electronic conductivity in terms of tunneling effect, i.e., the transmission of electrons among “disconnected” but close enough conductive nanoparticles, which takes place when the electrons actually “jump” from one GONS to another, bypassing the energy barrier opposed by the interposed cementitious matrix, and contact conduction, i.e., electronic/hole conduction through conductive paths formed by connected GONS [11,32]. For the extra-low dosage samples 0.45GO-0.01 and 0.55GO-0.01, the calculated ϕ_v_ is approximately 0.005% (Table 2), two orders of magnitude lower with respect to the percolation threshold (ϕ_p_). In fact, according to Garboczi et al. [33], ϕ_p_ can be evaluated considering the overlapping of 2D ellipsoids of width (W) and length (L). Herein investigated 2D nanofillers showed fully comparable values of W and L (Table 1), corresponding to a percolation threshold (ϕ_p_) equal to about 0.2%, expressed in terms of volume fraction percentage. More recently, several researchers assessed GNPs as a percolation threshold for GFCCs that lies between 1% and 5% by weight of cement [11].

Regarding samples 0.55GO-0.1 and 0.55GO-0.2, containing a much higher 0.55GO dosage, the ϕ_p_ threshold also abundantly exceeded the estimated ϕ_v_ values equal, respectively, to 0.5% and 0.1% (Table 2). Thus, since electronic conductivity has to be virtually excluded, it can be assumed that the electrical properties of all the investigated GO-engineered nanocomposites were dominated by ionic conductivity. Consequently, the observed trend of resistivity among the herein investigated samples might mainly lead back to a substantially different microstructural evolution due to the interaction of the highly oxidized GONSs with the hydrating cement matrix. At the nanoscale level (Level I) [34], the strong interfacial bonding between C-S-H and -COOH at the edges of GONSs results in a complex multiphase 3D framework characterized by enhanced cohesive forces and compactness [35]. As fully assessed by several studies, adding GONSs is expected to reduce porosity and refine the pore structure, the latter effectively impacting transport properties and, consequently, durability [19]. On this basis, in order to tentatively elucidate the resistivity trend observed by EIS measurements, the transport properties were investigated.

### 3.2. Permeability of GO-Engineered Mortars

A comprehensive permeability evaluation was performed on GO-modified mortars hardened at 28 days by measuring the volume percentage occupied by the permeable voids (VPV). The results indicated that the VPV value was reduced from 28.8% for CS to 27.4%, 27.1%, 27.4% and 27.0% for 0.55GO-0.01, 0.55GO-0.1, 0.55GO-0.2 and 0.45GO-0.01, respectively. Interestingly, a reduction of approximately up to 6% for all samples with respect to the control was found. Thus, the increase of 0.55GO dosage from 0.01% to 0.2% by weight of cement neither resulted in further reduced permeability (Table 2) nor improved mechanical strength according to the results. Conversely, despite the more favorable fresh and hardened properties of 0.45GO-0.01 compared to 0.55GO-0.01, these results suggest that the VPVs of the investigated samples were not affected by the oxidation degree of the GO nanosheets used in this investigation [13,14].

Aimed at elucidating the eventual role of the GO oxidation degree on the permeability of GO-nanocomposites, initial surface absorption tests (ISAT) and chloride ion diffusion tests were also performed on extra-low dosage 0.55GO-0.01 samples; the obtained results were compared to those previously reported for 0.45GO-0.01 (Figure 5a,b) [13].

Interestingly, when analyzing the ISAT results (Figure 5a), both extra-low dosage GO-modified nanocomposites showed comparable lower values with respect to the control, up to 55%. According to D. Dimov et al. [36], the enhanced formation of nucleation sites for the C-S-H hydration crystals on high-specific-surface-area 2D graphene oxide form a denser network of interlocked particles that acts as a water infiltration barrier, decreasing the amount of water that can penetrate the cementitious matrix through capillary pores and cracks. On the other hand, the results of the chloride ion diffusion test (Figure 5b) demonstrated the impact of the GO oxidation degree. The 045GO-0.01 sample clearly showed higher resistance to transport, not only with respect to the control sample as expected but also to the analog 0.55GO-0.01 sample. The transport properties are highly dependent on dosage, dispersion degree and thickness of GNDs due to the combination of refinement of capillary pores and the formation of a barrier due to enhanced tortuosity. For example, Mohammed et al. [37] suggested that the direct mixing of 0.01–0.06% by weight of cement of graphene oxide affects the microstructure of the cementitious matrix in terms of porosity and porous interconnectivity and hinders the penetration of chloride ions. Interestingly, these authors obtained the best performances for the same extra-low dosage samples herein considered (i.e., 0.01 wt% by weight of cement). According to the model proposed by Du et al. [38] for randomly distributed GNSs in a cement matrix, the tortuosity factor (τ) can be estimated according to the following:(10)$$\tau =1+\left(\frac{L}{t}\right)\left(\frac{\phi_{V}}{6}\right)$$
where ϕ_v_ is the volume fraction percentage, t is thickness and L is the length of the 2-D nanosheets. Thus, according to data reported in Table 1 and Table 2, the estimated τ values of 0.45GO-0.01 and 0.55GO-0.01 samples are both approximately equal to 1.0030. In order to possibly explain the observed difference of transport properties between 0.45GO-0.01 and 0.55GO-0.01 in terms of tortuosity factor, it should be considered that the available ϕ_v_ was significantly reduced due to the high agglomeration tendency of 0.55GO [14]. Samples 0.55GO-0.1 and 0.55GO-0.2 were characterized by higher τ values of 1.030(3) and 1.060(5), respectively. In conclusion, GO-modified modified mortars clearly showed, as expected, enhanced impermeability to water and chloride ions with respect to the control. However, since an exact match with the observed trend of the electrical resistivity of the sample cured at 28 days was not achieved (Table 3), a refined investigation on micrometric porosity was carried out by micro-computed tomography (μ-CT) analysis to clarify this point (see Section 3.4).

### 3.3. Thermal Conductivity of GO-Engineered Mortars

Thermal conductivity measurements were performed on the extra-low dosage GO-engineered nanocomposites, and the results are presented in Figure 6.

The results revealed that the increase in thermal conductivity (+57% compared to the control sample) was better fulfilled by 0.45GO-0.01. Since the 0.55GO and 0.45GO widths were approximately equal (Table 1), this result could be explained by assuming that 0.55GO-0.01 had a high agglomeration tendency [14] and lower ϕ_v_. Alternatively, the thermal conductivity of 55GO-0.01 might be driven by adverse microstructural features. Accordingly, G. Jing et al. [12] reported that mortars loaded with 1.2% (by weight of cement) of reduced graphene oxide (rGO) (O/C 0.036 atomic ratio) and dispersed in water with poly-carboxylate (PCE) exhibited a 7.8% increase in their thermal conductivity. It has been demonstrated by molecular dynamics (MO) simulations that orthotropic thermal behavior characterized graphene sheets [39]; at room temperature for a monolayer, the thermal conductivity ranges between 4840 W/(m K) and 5300 W/(m K) [40]. In a single GO nanosheet, epoxy and hydroxyl functional groups are non-uniformly distributed on the two sides of the carbon backbone. These functional groups adversely affect the thermal conductivity associated with the reduced phonon mean free path. Thus, higher-oxidation-level GO nanosheets showed reduced thermal conductivity due to enhanced phonon-defect scattering. It has been calculated that the intrinsic thermal conductivity at room temperature for a single layer of pristine graphene is around 2480 W/(m K), reduced to 72 W/(m K) for 0.35 oxidation degree [41]. The thermal properties of functionalized graphene depend heavily on the nanosheet length, especially for high oxidation levels, compared to pristine graphene sheets. Very low thermal conductivity is thus expected for long and highly oxidized graphene oxide nanosheets [41]. Regarding concrete/graphene nanocomposites, M. Ahmadi et al. [39] demonstrated with the finite element (FE) method that the improvement of thermal conductivity is directly related to GNDs in terms of distribution pattern, width and ϕ_v_. To be specific, the most significant increase occurs for the regular dispersion of graphene nanosheets along the direction of the thermal flow followed by the random dispersion pattern. The effect of regular dispersion perpendicular to the thermal gradient has to be considered negligible.

In order to focus on the correlation between thermal conductivity and microstructure, in terms of the amount and distribution of micrometric porosity, micro-computed tomography (μ-CT) analysis was performed on selected samples (discussed in the next section).

### 3.4. Micrometric Porosity of GO-Engineered Mortars

Cement-based materials are characterized by a complex arrangement of a multiphase, multiscale and porous structure that can be subdivided into four elementary levels ranging from the nanoscale (10^−8^–10^−6^ m, Level I) to the macroscale (10^−2^–10^−1^ m, Level IV). In this perspective, micrometric porosity is allocated in Level II (10^−6^–10^−4^ m) [34]. On this basis, to assess the extent of the impact of GO oxidation degree on micrometric porosity, μ-CT analysis was performed on extra-low dosage samples loaded with 0.45GO or 0.55GO (i.e., 0.45GO-0.01 and 0.55GO-0.01). Representative μ-CT cross-sections of samples and a binarized image of the control are shown in Figure 7. The summary results of 3D individual objects analysis (i.e., pores) are reported in Table 4. For every single object, the size is given in terms of a structure thickness value that represents the mean value of the particle distribution that fits the pore structure giving an estimate of its size; only for a spherical object, such value corresponds to pore diameter. The numeral and volumetric pore size distributions are shown in Figure 8 and Figure 9 in which the data are presented using points instead of the common histogram bars to make a clearer reading. The horizontal axis values are the mid value of each class range in the distribution, and to enhance the distribution features, labels on some data points were added. The overall mean size values are reported in Table 4. The micrometric porosity distributions normalized to CS are shown in Figure 10. Such results reflect the microstructural features evidenced in the 3D-reconstructed structures (Figure 11, Appendix A). The investigated GO-modified samples showed increased micrometric porosity with respect to the control; the values determined on 6 cm^3^ specimen for CS, 0.45GO-0.01, and 0.55GO-0.01 resulted in 1.97%, 2.11% and 3.0%, respectively. Thus, the effect was much more pronounced for the 0.55GO sample doped with GO of a higher oxidation degree. According to Zhang et al. [42], such porosity’s extent and size have to be associated with increased closed voids due to air trapping during the mixing phase. It is worth mentioning that these findings are also fully in line with previously investigated fresh properties of the admixtures whose plastic viscosity, workability and flowability trends follow the same trend [13,14].

Moreover, the results of μ-CT investigation also focus on several GO-engineered cementitious materials’ properties in relation to the oxidation degree of the 2D nanofiller. For example, according to Table 4, it is possible to assign the densification output of the GO nanocomposites to the interaction between GO and the cementitious matrix that can counterbalance the increased macroporosity induced by workability loss.

## 4. Conclusions

Two sets of cementitious nanocomposites were prepared using a commercially available premixed dry mortar (class M5) and a graphene oxide free-flowing nanopowder (Aldrich, oxidation degree 0.45, named 0.45GO) and a graphene oxide water suspension (Nanesa, oxidation degree 0.55, named 0.55GO). The dosage of the 2D nanofillers ranged from 0.01% to 0.2% (by weight of cement), corresponding approximately to a volume fraction percentage (ϕ_v_) within 0.005% and 0.1%. The samples were hardened in water at RT for 7, 14, and 28 days. The properties of the fresh mixtures as well as the mechanical properties of the respective hardened samples have been previously investigated. The proposed experimental study aimed to evaluate the impact of GO oxidation degree on the physical properties of the resulting cement-based nanocomposites. The results reported in this study clearly demonstrated that (i) the resistivity of cementitious composites depends not only as expected on curing time and dosage, but also on the oxidation degree of GO; (ii) thermal conductivity and resistance to chloride ions penetration were greatly favored by lower oxidation degree GO; (iii) micrometric porosity and pore density increased in nanocomposites loaded with higher oxidation degree GO; (iv) water permeability was reduced by the presence of GO, independently on the oxidation degree of the nanofiller. The physical and mechanical properties of cement-based nanocomposites are unavoidably related to the development of specific microstructural features, generally driven by the dispersion efficiency of the nanofiller within the cementitious matrix. In the Ca^+2^-rich alkaline environment typical of hydrating cement granules, graphene oxide has a particular tendency to agglomerate by Ca^+2^-mediated mechanisms that often result in unfavorable microstructures characterized by large and/or heterogeneous volume defects. This study, with the previous investigations of the group [13,14], demonstrates that employing an extra-low dosage of as-received GO nanosheets characterized by a moderately high oxidation degree is a practical approach to achieving cement-matrix materials with improved mechanical and physical properties.

## Figures and Tables

**Figure 1 nanomaterials-13-00726-f001:**
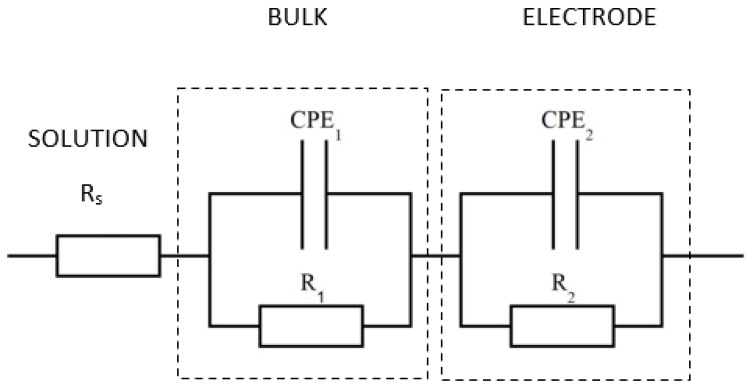
Equivalent circuit used to fit the impedance data.

**Figure 2 nanomaterials-13-00726-f002:**
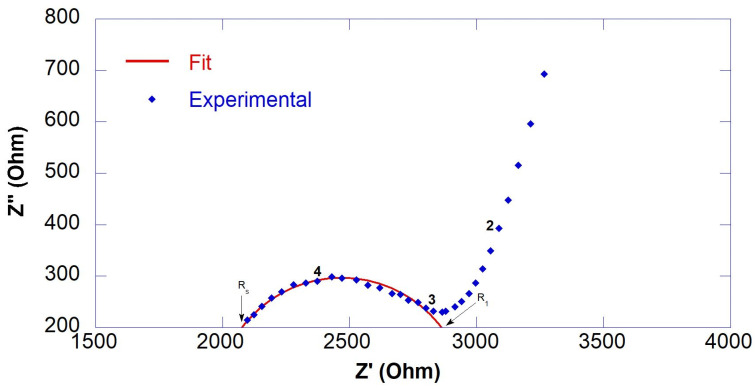
Experimental (marker) and fitted (line) EIS data of 0.55GO-0.01@28 days.

**Figure 3 nanomaterials-13-00726-f003:**
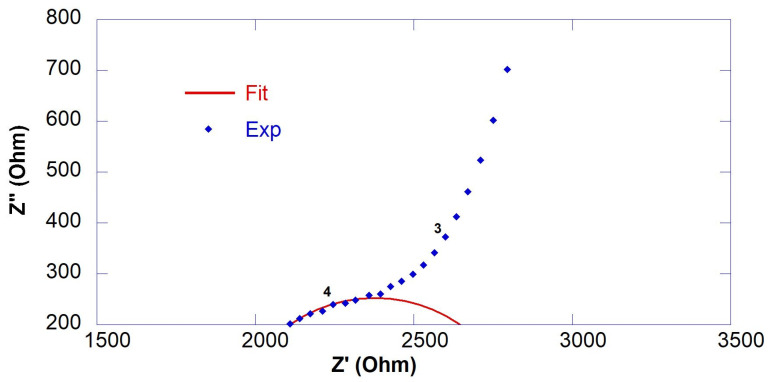
Experimental (marker) and fitted (line) EIS data of 0.55GO-0.1@28 days.

**Figure 4 nanomaterials-13-00726-f004:**
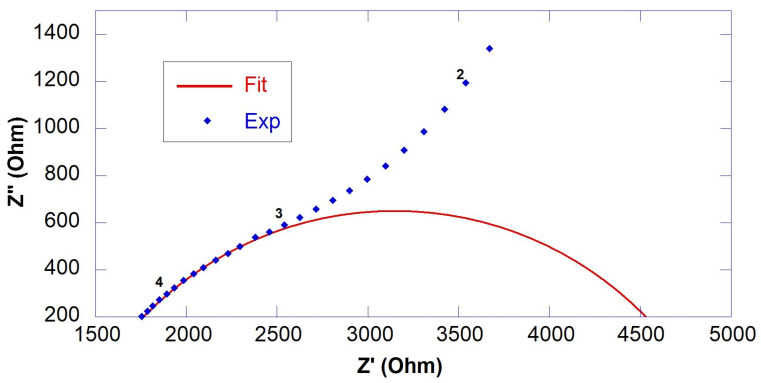
Experimental (marker) and fitted (line) EIS data of 0.55GO-0.2@28 days.

**Figure 5 nanomaterials-13-00726-f005:**
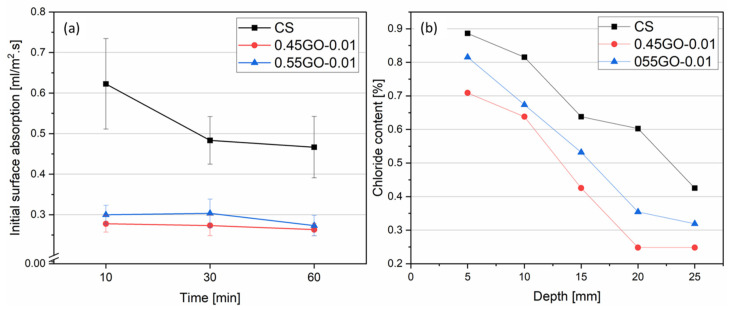
(**a**) Initial Surface Absorption Tests (ISAT) for 0.55GO-0.01 and 0.45GO-0.01@28 days, (**b**) Chloride ion diffusion at different depths for 0.55GO-0.01 and 0.45GO-0.01@28 days.

**Figure 6 nanomaterials-13-00726-f006:**
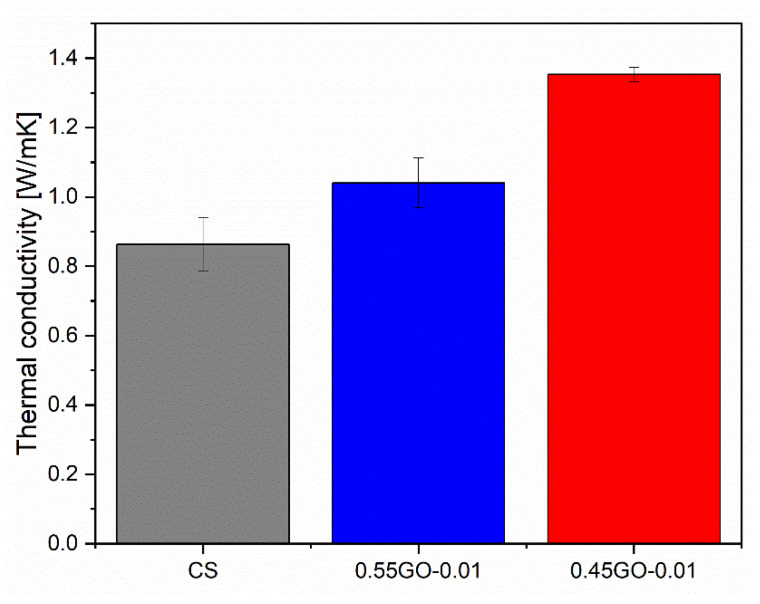
Thermal conductivity of 0.45GO-0.01 and 0.55GO-0.01 cured @28 days.

**Figure 7 nanomaterials-13-00726-f007:**
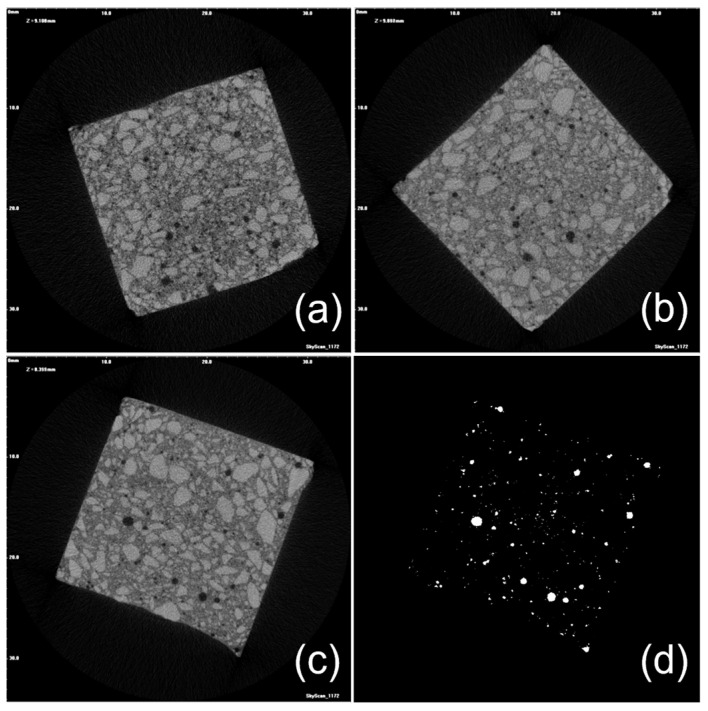
Representative μ-CT cross-sections of (**a**) 0.45GO-0.01, (**b**) 0.55GO-0.01, (**c**) CS and (**d**) CS binarized image.

**Figure 8 nanomaterials-13-00726-f008:**
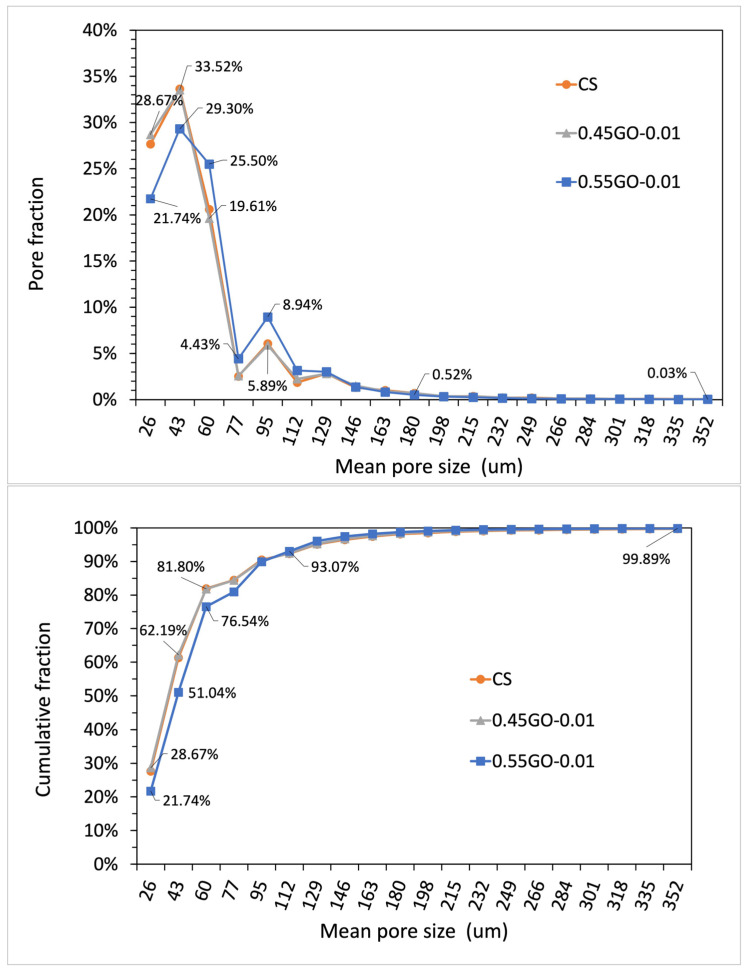
Numeral distribution of micrometric pores in terms of porosity fraction (**upper** panel) and cumulative porosity (**lower** panel) determined by μ-CT measurements of samples cured for 28 days (1 structure thickness = pixel resolution = 17.185 μm).

**Figure 9 nanomaterials-13-00726-f009:**
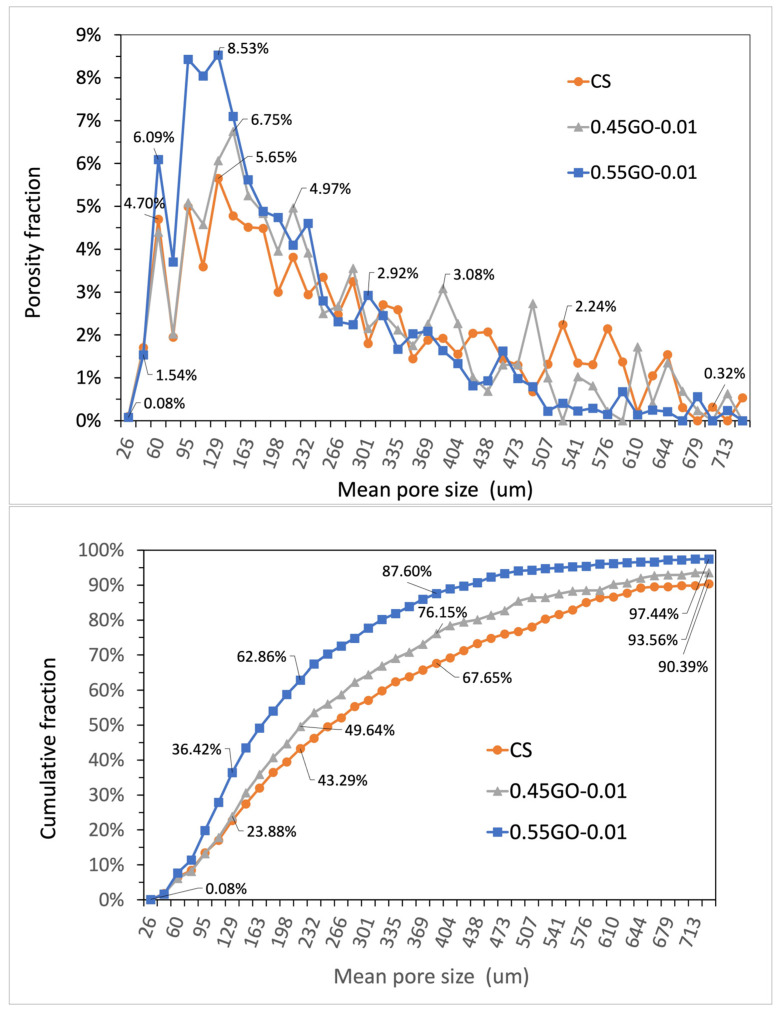
Volumetric distribution of micrometric pores in terms of porosity fraction (**upper** panel) and cumulative porosity (**lower** panel) determined by μ-CT measurements of samples cured for 28 days (1 structure thickness = pixel resolution = 17.185 μm).

**Figure 10 nanomaterials-13-00726-f010:**
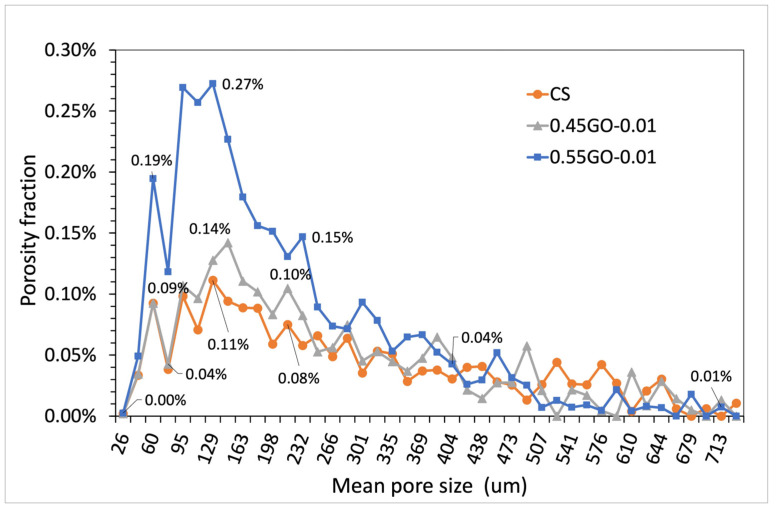
Volume fraction per unit volume of micrometric pores as determined by μ-CT measurements on nanocomposites cured for 28 days (1 structure thickness = pixel resolution = 17.185 μm).

**Figure 11 nanomaterials-13-00726-f011:**
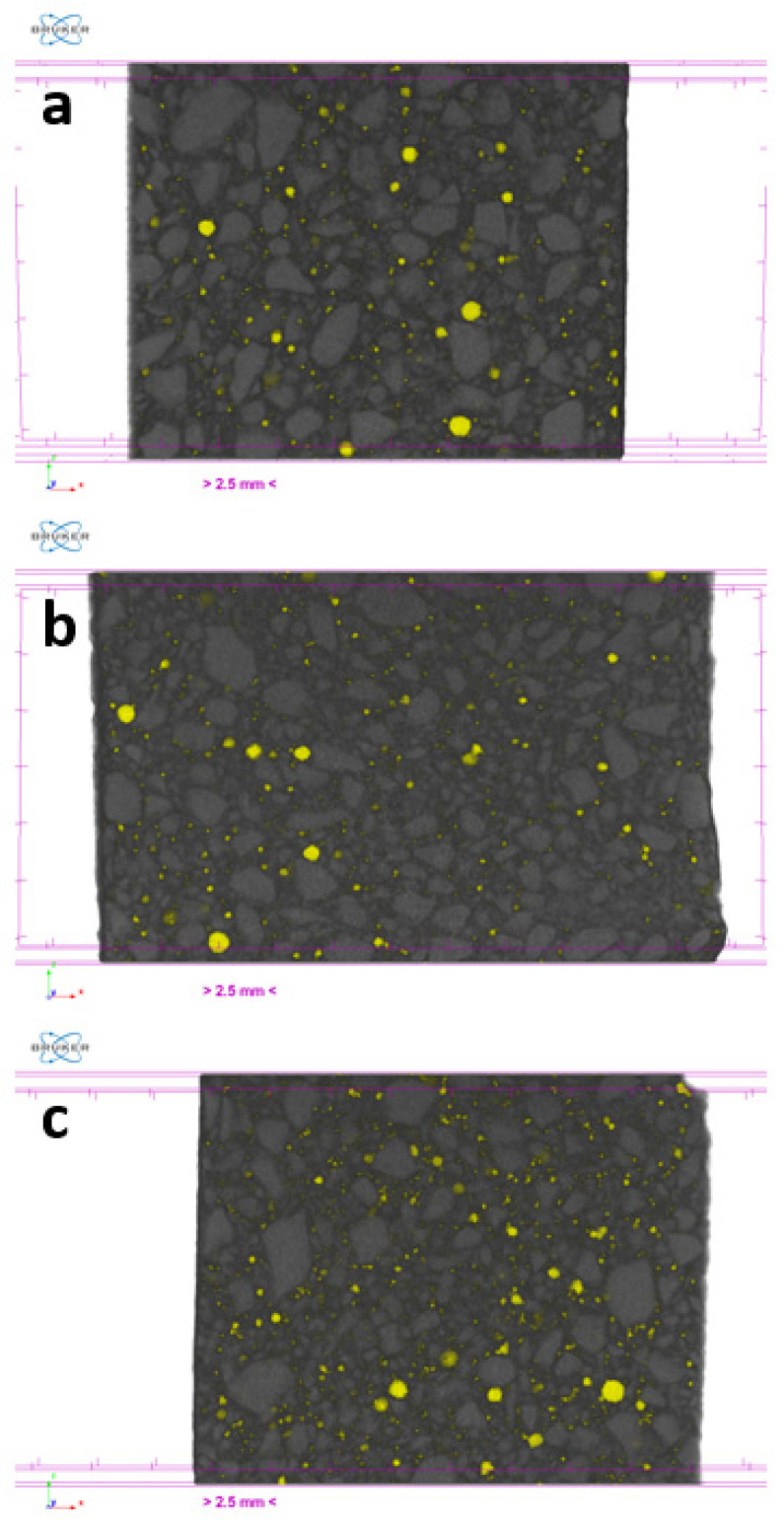
Thin sections (slices) of 3D reconstructed volume of (**a**) CS@28 days, (**b**) 0.45GO-0.01@28 days and (**c**) 0.55GO-0.01@28 days.

**Table 1 nanomaterials-13-00726-t001:** The designation, oxidation degree (molar ratio), size (maximum width and maximum length), thickness, aspect ratio, and interlayer spacing of the employed graphene oxide (GO).

Nanofiller	Oxidation Degree (O/C)	Designation	Width (W)	Length (L)	Thickness (t)	Aspect Ratio (Average Lateral Size/t)	Interlayer Spacing (t′)
GO (Aldrich)	0.45	0.45GO	600 nm	800 nm	2 nm	350	0.7835 nm
GONan (Nanesa)	0.55	0.55GO	600 nm	800 nm	2 nm	350	0.7795 nm

**Table 2 nanomaterials-13-00726-t002:** Formulation and designation of GO-engineered mortars.

Sample	Graphene Oxide (GO)	Weight Percentage (by Weight of Cement)	Weight Fraction Percentage (W_p_) 10^−3^	Volume Fraction Percentage @28 Days (ϕ_v_) 10^−3^	^a^ Density @28 Days g/cm^3^
Control Sample (CS)	-	-	-	-	1.945 ± 0.063
0.45GO-0.01	nanopowder	0.01%	2.29%	4.56% ^b^	2.210 ± 0.038
	(Aldrich)			5.40% ^c^	
0.55GO-0.01	suspension	0.01%	2.29%	4.54% ^b^	2.178 ± 0.043
	(Nanesa)			5.37% ^c^	
0.55GO-0.1	suspension	0.1%	22.90%	45.40% ^b^	2.176 ± 0.029
	(Nanesa)			53.7% ^c^	
0.55GO-0.2	suspension	0.2%	45.74%	90.68% ^b^	2.155 ± 0.310
	(Nanesa)			107.5% ^c^	

^a^ [13,14]; ^b^ calculated using as d_m_ the density value of CS@28days; ^c^ calculated based on the bar volume 4 × 4 × 16 cm (256 cm^3^) [14].

**Table 3 nanomaterials-13-00726-t003:** Results of EIS measurements: fitting parameters and resistivity (ρ ) of GO-engineered mortars at 100% RH and room temperature.

Sample	R_S_ (Ohm)	R_1_ (Ohm)	R_ccp_ (Ohm)	ρ (Ohm∙m)
CS@7days CS@14days CS@28days	- - 1836 ± 47	- - 2874 ± 200	* 1485 * 2104 4710	14.9 21.0 47.1
0.45GO-0.01@7days 0.45GO-0.01@14days 0.45GO-0.01@28days	- 1632 ± 54 1680 ± 16	- 993 ± 178 1846 ± 104	* 1624 2625 3526	16.2 26.2 35.3
0.55GO-0.01@7days 0.55GO-0.01@14days 0.55GO-0.01@28days	- - 1811 ± 65	- - 1651 ± 113	* 1541 * 3760 3462	15.4 37.6 34.6
0.55GO-0.1@7days 0.55GO-0.1@14days 0.55GO-0.1@28days	- 1647 ± 44 1743 ± 11	- 2366 ± 657 1691 ± 142	* 1657 4013 3434	16.6 40.1 34.3
0.55GO-0.2@7days 0.55GO-0.2@14days 0.55GO-0.2@28days	1275 ± 62 1584 ± 28 1545 ± 27	1210 ± 94 3734 ± 258 3122 ± 225	2485 5318 4667	24.8 53.2 46.7

* R_s_ and R_1_ not determined. R_ccp_ estimated by the fitting intercept.

**Table 4 nanomaterials-13-00726-t004:** Results of μ-CT analysis performed on GO-modified mortars cured for 28 days compared to the control.

Sample Density (g/cm^3^) ^a^	Volume of Interest (VOI) (mm^3^)	Porosity (%)	Total Volume of Micrometric Pores (mm^3^)	Number of Micrometric Pores * Pore Density (pores/mm^3^)	Mean micrometric Pore Size (Numeral) * St. Dev. (μm)	Mean Micrometric Pore Size (Volumetric) * St. Dev. (μm)
CS (1.945 ± 0.063)	6482	1.97	127	59265 * 9.1	57.3 * 45.9	337.5 * 249.2
0.45GO-0.01 (2.210 ± 0.038)	6864	2.11	145	65835 * 9.6	56.5 * 42.7	300.9 * 237.9
0.55GO-0.01 (2.178 ± 0.043)	6553	3.20	209	95332 * 14.5	60.2 * 38.7	233.8 * 252.8

^a^ [13,14].

## Data Availability

Not applicable.

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
