# Peer review of "An Insight into Durability, Electrical Properties and Thermal Behavior of Cementitious Materials Engineered with Graphene Oxide: Does the Oxidation Degree Matter?"

_nanomaterials, 2023, doi:10.3390/nano13040726_

Round 1

Reviewer 1 Report

The paper mainly focuses on the effect of two distinct GONSs on durability, electrical properties and thermal behavior of cement mortars from experimental point of view. Electrochemical Impedance Spectroscopy and Heat Flow Meter tests were conducted to investigate the electrical and thermal properties, and the test results were discussed with regarding to microscopic pore characteristics and macroscopic transport properties. I feel the manuscript deserves publication if the following comments are detailly explained or corrected by the authors:

1.     As is known, the improvement of GO nanosheets on cementitious materials is closely related to its dispersibility in the cement matrix. However, there is no data on the dispersibility of GONSs, especially in case that two GO products (nano powder and water suspension) were used in this study. Therefore, nano dispersibility which is vital to the behaviors of GO-cement composites should be explicitly explained. Although systematic characterization of both nanofillers has been determined in previous published articles, the conclusion may be unreliable only based on these data and more information is still essential for this study as:

1) How to mix GO products with available premixed dry mortar? Hand stirring or ultrasonic treatment?

2) How to evaluate dispersibility of the two nanofillers in the cement matrix respectively?

3) General speaking, GO aqueous solution is more easily dispersed into the cement matrix, while GO powder is almost impossible for to disperse evenly and tends to agglomerate together due to the strong intermolecular force. How to ensure the reliability of your experimental results? The suitable tests or explanations are suggested in the article.

2.  In table 2, “weight percentage” donates the weight ratio of GO powder to cement. For GO water suspension, “weight percentage” also means the weight ratio of water suspension to cement? If yes, please explain how the water (from GO water suspension) may affect the experimental results? Whether this part of water will change the water to cement ratio and thus weaken the behaviors of mortars?

3.  If possibility, the information of two GO should be provided including the surface functional groups which will be helpful for understanding the enhancing mechanism, the Raman spectrum which can be used to evaluate the quality of GO.

4.  Two GO products have been used in this study, while their performance such as oxidation degrees, aspect ratio, mechanical property, morphology and so on are different from each other. Why only oxidation degrees (oxygen to carbon molar ratios are 0.55 and 0.45 respectively) are highlight in the Abstract?

5.  The data figure in the article is too unrefined. As shown in Figure 5, Figure 8- Figure 10, the label of the first data point or the last data point cannot be displayed completely, please correct. The author should adjust the horizontal axis slightly and make appropriate adjustments to make the picture beautiful.

6.  In Conclusions (Page 17), it seems to be a list of experimental results. Please further refine the summary and rewrite this part.

Reviewer 2 Report

Very well written manuscript reporting interesting results. Few minor corrections are suggested.

1) Table 2 is scattered and needs to be edited.

2) Page 4, line 157: The authors state "...Measurements were collected at room temperature at  100 RH by applying ...". RH probably stands for relative humidity. The authors should clarify this. Also it would be better to write 100%RH.

3) Page 7, lines 318 and 319: It is exaggeration to  report VPV values on 3 decimals; reporting values on one decimal would be more appropriate and likely more in agreement with actual precision of the method. 

4) Page 10, line 329: Space between "...0.55GO-0.01 and samples; ... ". 

5) Page 18, line 530: Year of publishing, i.e. 2015 should be bold. 

Round 2

Reviewer 1 Report

All suggestions for authors have been well considered and revised.

Author Response

All suggestions for authors have been well considered and revised.

R: Thanks